# NAMPT Inhibitor and P73 Activator Represses P53 R175H Mutated HNSCC Cell Proliferation in a Synergistic Manner

**DOI:** 10.3390/biom12030438

**Published:** 2022-03-12

**Authors:** Bi-He Cai, Zhi-Yu Bai, Ching-Feng Lien, Si-Jie Yu, Rui-Yu Lu, Ming-Han Wu, Wei-Chen Wu, Chia-Chi Chen, Yi-Chiang Hsu

**Affiliations:** 1School of Medicine, I-Shou University, Kaohsiung 82445, Taiwan; lien980206@yahoo.com.tw (C.-F.L.); danielwu0918@gmail.com (M.-H.W.); 2Department of Medical Laboratory Science, I-Shou University, Kaohsiung 82445, Taiwan; baizhizhi2002@gmail.com (Z.-Y.B.); twruuq@gmail.com (S.-J.Y.); evafen88@gmail.com (R.-Y.L.); 3Department of Otolaryngology-Head and Neck Surgery, E-DA Hospital, Kaohsiung 82445, Taiwan; 4Department of Physical Therapy, I-Shou University, Kaohsiung 82445, Taiwan; rise00721@gmail.com; 5Department of Pathology, E-Da Hospital, I-Shou University, Kaohsiung 82445, Taiwan

**Keywords:** p53, p73, hand and neck, NAMPT, aggregation

## Abstract

The p53 family has the following three members: p53, p63 and p73. p53 is a tumor suppressor gene that frequently exhibits mutation in head and neck cancer. Most p53 mutants are loss-of-function (LoF) mutants, but some acquire some oncogenic function, such as gain of function (GoF). It is known that the aggregation of mutant p53 can induce p53 GoF. The p73 activators RETRA and NSC59984 have an anti-cancer effect in p53 mutation cells, but we found that p73 activators were not effective in all head and neck squamous cell carcinoma (HNSCC) cell lines, with different p53 mutants. A comparison of the gene expression profiles of several regulator(s) in mutant HNSCC cells with or without aggregation of p53 revealed that nicotinamide phosphoribosyltransferase (NAMPT) is a key regulator of mutant p53 aggregation. An NAMPT inhibitor, to reduce abnormal aggregation of mutant p53, used in combination with a p73 activator, was able to effectively repress growth in HNSCC cells with p53 GoF mutants. This study, therefore, suggests a potential combination therapy approach for HNSCC with a p53 GoF mutation.

## 1. Introduction

The p53 family has the following three members: p53, p63 and p73 [1,2]. p53 is a tumor suppressor gene that frequently exhibits mutation in cancers [3]. Sixty-three percent of head and neck squamous cell carcinomas (HNSCC) contain p53 mutations, which is above the average of close to 50% for all types of cancer [4,5]. Previously, studies mostly focused on the loss of function (LoF) of the tumor suppressor role of wild-type p53, due to mutation, but, recently, there has been more focus on the gain of function (GoF) in mutant p53 proteins [5]. The acquisition of p53 GoF after a mutation is currently considered to occur via three mechanisms [6,7]. In the first, the mutant p53 directly binds to the novel binding site with a non-canonical sequence to activate a specific downstream gene [8]. In the second, the mutant p53 acts as a co-activator to bind to a specific transcription factor to activate the downstream gene [9]. In the third, the mutant p53 binds to a specific transcription factor to cause it to lose its transcription ability [10]. It is known that the aggregation of mutant p53 can induce p53 GoF through sequestration of other tumor suppressor genes; thus, such aggregation is an example of the third mechanism [11,12,13].

The p73 activators RETRA and NSC59984 have an anti-cancer effect in p53-mutated cells [14,15]. The molecular mechanism of these two drugs is to block the mutant p53 and p73 interaction, or to promote mutant p53 for degradation to reactivate p73 [14,15], so the amounts of p73 show no large changes, but it can be released from mutant p53. It, thus, makes sense that these two p73 activators have no anti-cancer effect in p53 null or p53 wild-type cells [14,15]. Aggregative-type p53 mutants can pull down the other tumor suppressor genes, such as p63 and p73, resulting in LoF [13,16]. So, p73 activators may have a different anti-cancer effect in non-aggregative and aggregative p53 mutants.

In this study, we compared two p53 mutant HNSCC cell lines, OECM1 and Detroit 562 cells, with different p53 mutations, and treated them with p73 activator(s). The Detroit 562 cell line has a p53 R175H mutation [17], and OECM-1 has a p53 V173L mutation [18]. We found that both RETRA and NSC59984 had a better cytotoxic response in OECM1 cells than in Detroit 562 cells. We evaluated whether the p53 R175H and V173L mutations were prone to aggregation or not, using the CSSP method [19], and assayed the aggregative signal in these two cell lines. After assaying the key molecules that influence the aggregative signal, and comparing the gene expression profiles of these two cell lines, we found that nicotinamide phosphoribosyltransferase (NAMPT) is a good target to repress aggregative signals in cells. We further treated Detroit 562 cells with a high aggregative signal with an NAMPT inhibitor and either one of the p73 activators, and found that an NAMPT inhibitor and a p73 activator have synergic effects in repressing Detroit 562 cell proliferation. This study may, thus, provide a potential treatment strategy for malignant HNSCC tumors with p53 GoF mutants.

## 2. Materials and Methods

### 2.1. Cell Culture and Drug Treatment

The HNSCC cell lines Detroit 562 and OECM-1 were maintained at 37 °C in 5% CO_2_ in MEM (for Detroit 562) or RPMI 1640 (for OECM-1) (Invitrogen, Carlsbad, CA, USA), supplemented with 10% FBS (Invitrogen), 100 U/mL penicillin and 100 μg/mL streptomycin (both from Invitrogen). HNSCC cells were treated with mock (DMSO only), 40 μM RETRA (a p73 activator, Santa Cruz, CA, USA), 100 μM NSC59984 (a p73 activator, Sigma-Aldrich, St. Louis, MO, USA), 10 μM EX527 (a SIRT1 inhibitor, Sigma-Aldrich, St. Louis, MO, USA), 10 μM VER-155008 (an HSP70 inhibitor, MedChemExpress, Monmouth Junction, NJ, USA) or 10 μM FK886 (an NAMPT inhibitor, MedChemExpress, Monmouth Junction, NJ, USA).

### 2.2. CCK8 Assay

Cell viability was assessed using CCK8 assay (Invitrogen). Mock was calculated as 100% to normalize for other drug treatment conditions. CCK8 (10 μL) was added to each 96 well, and the plate was put in an incubator at 37 °C with 5% CO_2_ for 1 h; then, the OD 450 nm was read on a microplate reader SpectraMax iD3 (Molecular Devices, Silicon Valley, CA, USA). Medium only was used as the blank, and the percentage of cell viability was calculated as follows: (Drug OD-Blank OD/Mock OD-Blank OD) × 100%.

### 2.3. Thioflavin T Staining

Thioflavin T (Sigma-Aldrich) stock solution (10 mM in water) was mixed with 1X dPBS to obtain a final working concentration of 10 μM (1:1000). Hoechst 33342 (AAT Bioquest, Sunnyvale, CA, USA) stock solution (1 mg/mL) was mixed with 1X dPBS to obtain a final working concentration of 1 μg/mL (1:1000). After co-staining both ThT and Hoechst 33342 for 30 min, the attached cells were washed with 1X dPBS three times. The ThT signal was detected at the excitation wavelength of 450 nm and an emission wavelength of 490 nm on a microplate reader SpectraMax iD3 (Molecular Devices). The Hoechst 33342 signal was detected at the excitation wavelength of 360 nm and the emission wavelength of 460 nm on a microplate reader. The protein aggregative signal was calculated as the OD ratio of ThT/Hoechst 33342 as normalized ThT fluorescence intensity.

### 2.4. Immunocytochemistry

The culture media was removed from each 96 well and washed twice with 1X PBS. Formaldehyde fixative solution (100 μL; 4% paraformaldehyde in 1X PBS) was added to each well, and incubated for 20 min. The wells were washed twice with 100 µL of wash buffer (1% BSA in 1X PBS). Blocking buffer (100 µL; 1% BSA, 0.2% Triton X-100 in 1X PBS) was added and incubated for 30 min, then the blocking buffer was removed. Primary antibody p53 DO-1 (1 μL; Santa Cruz; sc-126) (1:100) was added to 100 µL blocking buffer to stain cells for 2 h. The wells were washed three times with 100 µL of wash buffer. Secondary antibody PE-conjugated mouse IgG2a (0.5 µL; 1:200), 1 μL of thioflavin T stock solution (10 mM) and Hoechst 33342 stock solution (1 mg/mL) were added to 100 µL blocking buffer to stain cells for 30 min in the dark. The wells were washed three times with 100 µL of wash buffer. Images were visualized and captured using a fluorescence microscope (ECLIPSE. Ts2; Nikon, Tokyo, JP), and merged images were created using ImageJ [20].

### 2.5. Real Time RT-PCR

Total cellular RNA was extracted with TRIzol reagent (Invitrogen). First-strand cDNA for mRNAs was prepared from 1 μg total RNA with the QuantiNova Reverse Transcription Kit (QIAGEN, Hilden, Germany). cDNA samples were mixed with primers and 5X HOT FIREPol EvaGreen qPCR Mix Plus (Omics Bio, New Taipei City, Taiwan). The amplification of all qPCRs was monitored using the QuantStudio 3 Real-Time PCR System (Applied Biosystems). Relative transcript levels were calculated as 2−ΔΔCT. The forward □ (F) □ and reverse (R) □ primers used were as follows: GAPDH, F: GTCTCCTCTGACTTCAACAGCG and R: ACCACCCTGTTGCTGTAGCCAA; NEU4, F: GCCTGAGGCCGTGCAGATCG and R: GCCGGGACCCACAGCGAATG; JAG2, F:□GAG CTG GAA CGA GAC GAG TG and R: TCC AGG TTA TAG CAG CGA GC; LIG1, F:□ TGC TTC CCT CTC TGA CAC CT and R: GAA TGG TCC GTT TCG GGA GC; G6PD, F:□GCC TTC CAT CAG TCG GAT ACA and R: CAC GAT GAA GGT GTT TTC GGG; NAMPT, □ F: TCC CAA GAG ACT GCT GGC AT and R: GGT CTT TCC CCC AAG CTG TTA; SIRT1, F: CCG AGA TAA CCT TCT GTT CGG T and R: CTA TCC GTG GCC TTG GAG TC;□HSP70, F: ACA TCA GCC AGA ACA AGC GA and R: AGT CGA TGC CCT CAA ACA GG.

### 2.6. Drug Interaction Analysis

The coefficient of drug interaction (CDI) was calculated as follows: CDI = AB/(A × B). According to the cell viability of each group from CCK8 assay, AB is the ratio of the combination drug group; A or B is the ratio of the single drug group. Thus, CDI value <1 = 1, or >1 indicates that the drugs are synergistic, additive or antagonistic, respectively. CDI <0.7 indicates that the drug is significantly synergistic.

## 3. Results

### 3.1. p73 Activators Have Different Responses in HNSCC Cell Lines with p53 V173L or p53 R175H

In this study, we found that two p73 activators, RETRA and NSC59984, have much higher cytotoxicity in OECM1 cells than in Detroit 562 cells (Figure 1). Detroit 562 cells have a p53 R175H mutation [17], and OECM-1 cells have a p53 V173L mutation [18]. The p73 specific downstream genes NEU4, JAG2, LIG1 and G6PD [21,22,23,24] were upregulated within RETRA- or NSC59984-treated OECM1 cells (Figure 2).

### 3.2. p53 V173L is A Non-Aggregative Type of p53 Mutant

It is known that p53 R175H is an aggregative type of p53 mutant, as a GoF-type mutation [25,26], but, to the best of our knowledge, there is no report on whether p53 V173L is prone to aggregation or not. Inclusion bodies, folding aggregates, and thermal aggregates acquire a new β-strand structure following an aggregative process involving intermolecular interactions [27]. NetCSSP is an online prediction tool, which provides a graphic interface that enables an interactive calculation of contact-dependent secondary structure propensity (CSSP) values for a given protein sequence [19]. The CSSP values of the β-strand in p53 V173L are lower than those in p53 R175H (Figure 3), so we predicted that OECM1 cells with p53 V173L should have fewer protein aggregative signals compared to Detroit 562 cells with a p53 R175H mutation. We used thioflavin T (ThT) to stain aggregative proteins in the cells, and found that the thioflavin T signals were colocalized with mutant p53 in the Detroit 562 cells (Figure 4A). Although most of the mutant p53 and thioflavin T signals are located in the nucleus, there are some cells with cytosolic staining of both mutant p53 and thioflavin T signals (Figure 4B). Wild-type p53 is localized in the nucleus to perform its biological function [28]. p53 R175H is found in both the nucleus and cytoplasm, although most cells were concentrated in the nuclei [29]. Detroit 562 cells had more protein aggregation signals than OECM-1 cells (Figure 5).

### 3.3. NAMPT Inhibitor and p73 Activator Have Synergic Effects That Repress the Growth of HNSCC Cells with p53 R175H Mutation

Heat shock protein 70 (HSP70) has been reported to enhance mutant p53 aggregation [30]. SIRT1 is a NAD+-dependent histone deacetylase that has been reported to deacetylate HSF1 to enhance HSF1 transcriptional activity and increase HSP70 expression [31].

NAMPT is a rate-limiting enzyme in the nicotinamide adenine dinucleotide (NAD+) salvage pathway, used to convert nicotinamide to nicotinamide mononucleotide (NMN), which is the immediate precursor of NAD+ formation in mammals [32]. NAMPT can enhance SIRT1 activity by increasing the amount of NAD+ [33]. We assayed the HSP70, SIRT1, and NAMPT expression in Detroit 562 and OECM1 cells, and found that all three genes have much higher expression in Detroit 562 cells than in OECM1 cells (Figure 6). NAMPT had significantly different expression in these two cell types, so we predicted that the NAMPT inhibitor may be effective in reducing mutant p53 aggregation. We found that the NAMPT inhibitor had a much greater effect on reducing the ThT signal than the SIRT1 inhibitor or HSP70 inhibitor (Figure 7). The NAMPT inhibitor and p73 activator repressed cell proliferation in Detroit 562 cells in a synergistic manner (Figure 8).

## 4. Discussion and Conclusions

Aggregation of mutant p53 can induce p53 GoF [11,12,13]. p53 mutant aggregation has been most often reported in breast, lung and ovary cancer cells [34,35,36]. To the best of our knowledge, only two articles have reported head and neck cancer; p53 mutant aggregation was reported in the HNSCC cell line Detroit 562 with a p53 R175H mutation [25,37]. There are several hotspot mutations in codons 175, 179, 196, 213, 220, 245, 248, 273 and 282 of p53 in HNSCC [38,39], and HNSCC can be divided into different subtypes [40,41]. More information about whether p53 hotspot mutations can become aggregated or not in HNSCC, and how aggregated mutant p53 obtains GoF function in different sub-types of HNSCC, will need to be further addressed in the future.

All the p53 family members have different isoforms, which can be divided into the following two groups: TA or ∆N isoforms [2]. ∆N isoforms act as the dominate-native function to interact with TA isoforms and block their transactivation function [42]. We did not assay the expression profile of each p63 and p73 isoform in our HNSCC cells in this study, so it is hard to say whether isoforms of p63 will influence the drug response(s) in our p73 activator combination with aggregation blocker(s). However, the α isoform is the major isoform in p63 and p73 [43]. p53 R175H can interact with ∆Np63α and ∆Np73α through transactivation inhibitory (TI) domains within the α isoform-specific C termini. The expression of TAp73 is higher than ∆Np73 in most organs [44], but the major p63 isoform is ∆Np63 in most organs, besides in the testis and skeletal muscle, which have high TAp63 [43]. This is why only the p73 activator has been developed [14,15], and no p63 activator has been developed to date. The relationship between ∆Np63α and p73 activator drug(s) responses in different p53 mutant HNSCC cell lines should be investigated in the near future. In addition, ankyloblepharon-ectodermal defects-cleft lip/palate (AEC) syndrome in the carboxyl (C) terminus of the ∆Np63α isoform with a mutation can form aggregation [45]. AEC-associated mutant p63 could co-aggregate with wild-type p63 and wild-type p73, but not with wild-type p53 [45]. AEC-associated mutant p63 exhibited impaired DNA binding and transcriptional activity as dominant negative effects, due to co-aggregation with wild-type p63 and p73 [45]. This may explain why AEC can cause severe skin fragility, but not induce any type of cancer formation. One ∆N isoform of p53, Δ133p53β, but not Δ133p53α or Δ160p53β, can form aggregation [46]. Δ133p53β aggregation is controlled by the CCT complex, but not HSP70, which regulates p53 R175H mutant aggregation [46]. Aggregative Δ133p53β can deactivate its dominate-native effect [46], thus aggregative Δ133p53β acts as a relatively beneficial aggregative compared to GoF mutant p53 or AEC-associated mutant p63, which have disease-type aggregation. Which kinds of protein can be pulled down by AEC-associated mutant p63 or Δ133p53β, and what kinds of factors will influence AEC-associated mutant p63 or Δ133p53β aggregation, will also be interesting topics for development biology and cancer biology in the future.

Several approaches have been taken to resolve the GoF p53 mutation’s oncogenic function [47]. The first is to reduce the amount of mutant p53; for example, MCB-613, a small-molecule stimulator of steroid receptor coactivators, can degrade p53 R175H through a USP15-dependent lysosomal pathway [48]. The second is to convert mutp53 to the wild-type p53-like structure; for example, PRIMA-1 and its methylated form PRIMA-1^Met^ (also called APR-246) can bind to p53 R175H to refold as the wild-type p53-like conformation. Mutant p53 can associate with p63 and p73, whereas wild-type p53 cannot [49,50], so p73 can also be reactivated by APR-246 [51]. The third is to block the interaction between p73 and mutant p53 to relieve normal p73 anti-cancer function, similarly to RETRA and prodigiosin [14,52]. Here, we provide new data, which are related to the third approach, and found that aggregative degrees of different p53 mutations can influence p73 activator function. The combination treatment with the NAMPT inhibitor could reduce the p53 R175H aggregative signal to refine the p73 activator effect. A similar combination treatment idea was also used recently to add APR-246 to reactive GoF p53 mutants, to refine the anti-cancer effect of berberine and modified berberine compounds in pancreatic cancer, according to the second approach as mentioned above [53].

The results reported herein show that V173L is a relatively non-aggregative p53 mutant compared to R175H in HNSCC cell lines. p73 activator(s) work well in V173L-containing cells, but have a poor response in R175H-containing cells. The NAMPT inhibitor is effective for reducing mutant p53 aggregation, and the NAMPT inhibitor and p73 activator, used as a co-treatment, act synergistically to repress cell proliferation in HNSCCs with the GoF p53 R175H mutant. This study, therefore, suggests a potential combination treatment strategy for HNSCC tumors with a p53 GoF mutation.

## Figures and Tables

**Figure 1 biomolecules-12-00438-f001:**
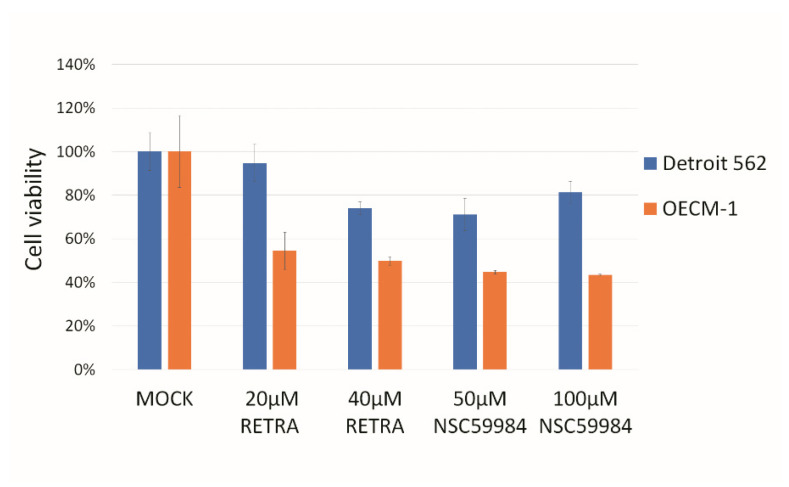
p73 activators can repress HNSCC growth in HNSCC cell lines. Two p73 activators, RETRA and NSC59984, were able to repress OECM-1 cell growth and, to a lesser extent, Detroit 562 cell growth. Mock was calculated as 100% to normalize for other conditions.

**Figure 2 biomolecules-12-00438-f002:**
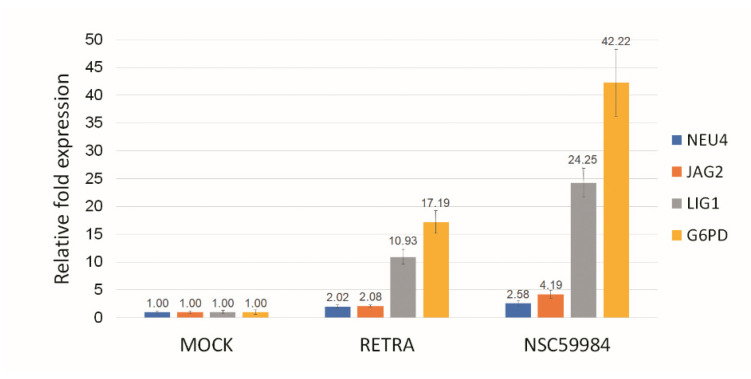
p73 activator(s) activate p73 downstream genes in OCEM-1 cells. RETRA (40 μM) or NSC59984 (100 μM) could upregulate p73 downstream genes NEU4, JAG2, LIG1 and G6PD in OECM-1 cells. Mock (DMSO only) was calculated as 1 to normalize for other conditions.

**Figure 3 biomolecules-12-00438-f003:**
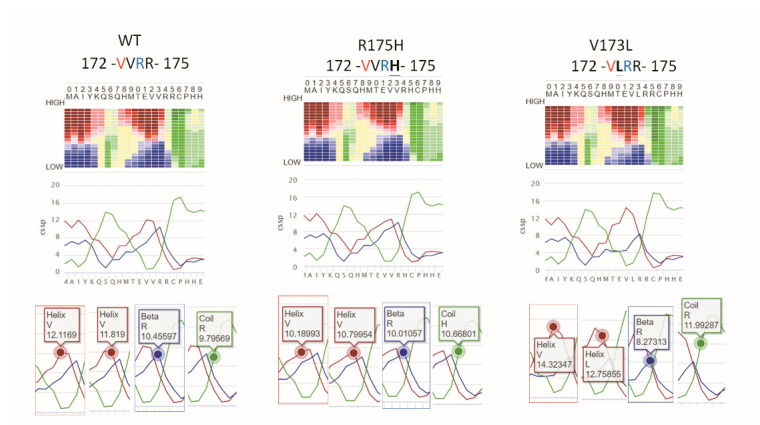
The CSSP value of the β-strand of flanking amino acids (a.a.) on p53 V173L is lower than the wild-type (WT) and R175H. The propensity for each of the three secondary structure elements, α helix (red), β-strand (blue) and coil (green), is calculated at 20 different levels, so the sum of the CSSP scores is 20. R175H flanking a.a. reduces α-strand CSSP from 12.1169 (WT) to 10.18993. V173L flanking a.a. reduces β-strand CSSP from 10.45597 (WT) to 8.27313. V173L flanking a.a. has a 14.32347 α-strand CSSP larger than WT (12.1169) and R175H (10.18993).

**Figure 4 biomolecules-12-00438-f004:**
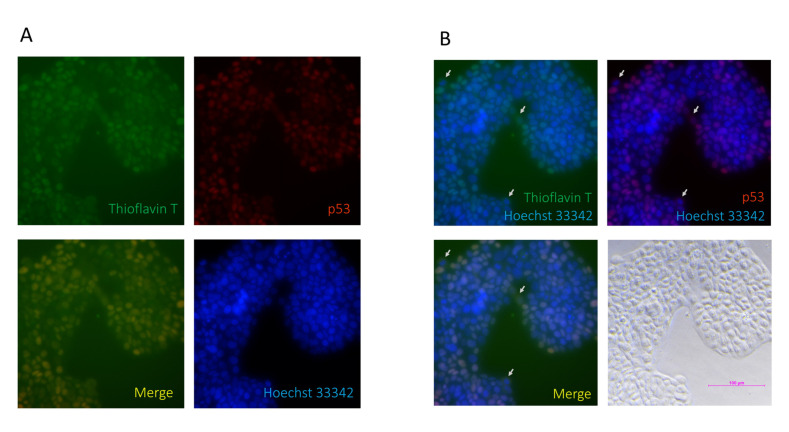
p53 is colocalized with thioflavin T in Detroit 562 cells. (**A**) p53 was used to stain for protein aggregation and is shown in red; thioflavin T was used to stain for protein aggregation and is shown in green; Hoechst 33342 was used to stain for nuclear counterstains and is shown in blue. p53 signals are shown to have the same location as the thioflavin T signals in the merged figure. (**B**) The merged figure shows that there are some cells with cytosolic thioflavin T/p53 signals (arrows).

**Figure 5 biomolecules-12-00438-f005:**
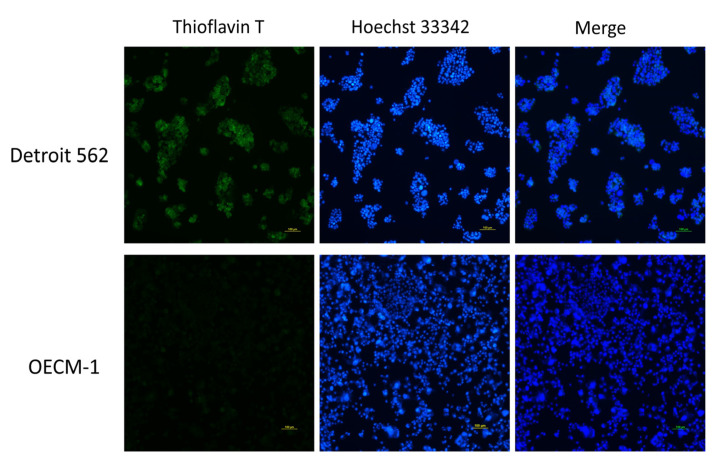
Protein aggregation in HNSCC cell lines. Thioflavin T was used to stain for protein aggregation and is shown in green; Hoechst 33342 was used to stain for nuclear counterstains and is shown in blue. Detroit 562 cells showed more thioflavin T signals than OECM1 cells.

**Figure 6 biomolecules-12-00438-f006:**
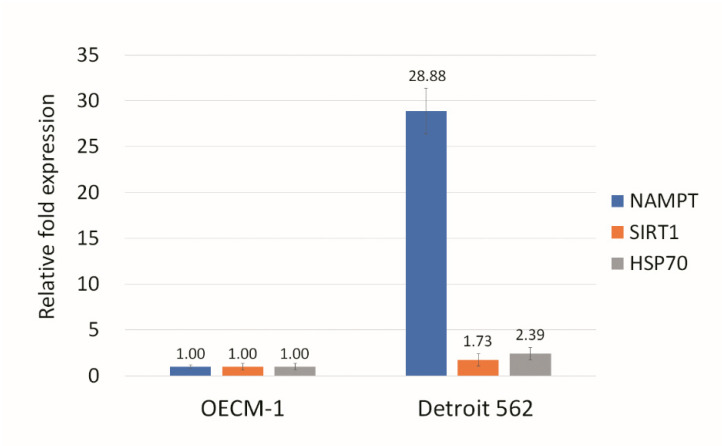
NAMPT has a higher level of expression in Detroit 562 cells than in OECM-1 cells. Although the levels of NAMPT, SIRT1, and HSP70 expression in Detroit 562 were larger than in OECM-1, the NAMPT expression showed, by far, the greatest difference between the two cell lines. Gene expression in OECM1 cells was calculated as 1.

**Figure 7 biomolecules-12-00438-f007:**
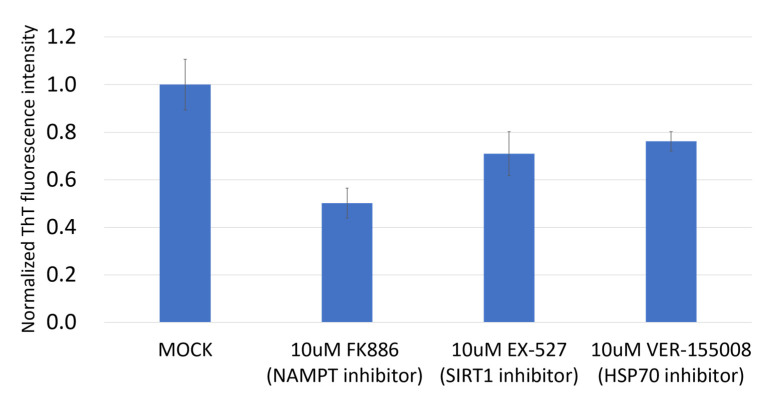
NAMPT inhibitor was able to repress protein aggregation signals. Although all the inhibitors tested (NAMPT, SIRT1 and HSP70) reduced the aggregation signals in Detroit 562, the NAMPT inhibitor had the most pronounced effect in reducing the protein aggregation. The protein aggregation signal was calculated as the OD ratio of ThT/Hoechst 33342 as normalized ThT fluorescence intensity. Mock was calculated as 1 to normalize for other conditions.

**Figure 8 biomolecules-12-00438-f008:**
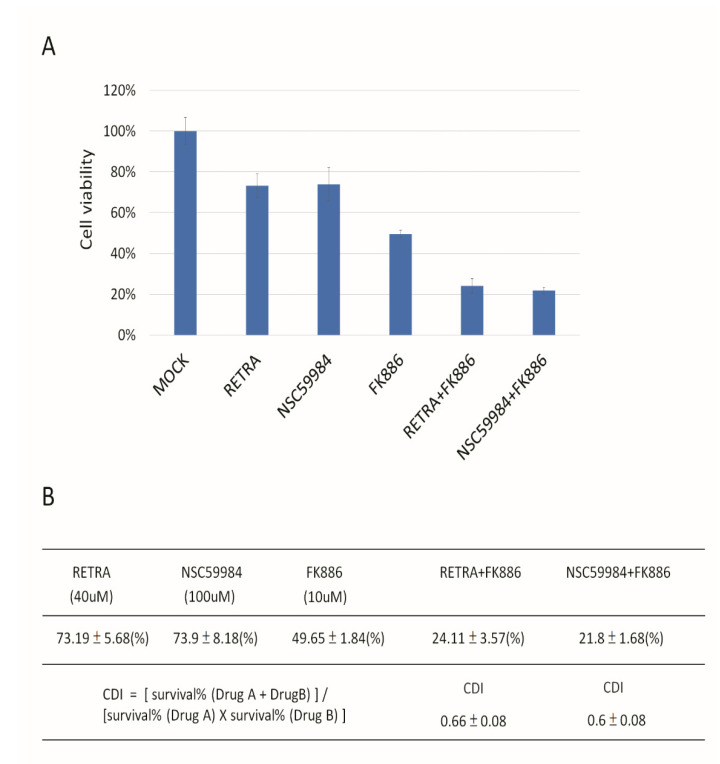
Synergistic repression of Detroit 562 cell growth mediated by p73 activators and NAMPT inhibitors. (**A**) RETRA, NSC59984 and NAMPT inhibitor (FK886) can inhibit Detroit 562 growth. Mock was calculated as 1 to normalize for other conditions. (**B**) The coefficient of drug interaction (CDI) was calculated as (A + B)/A×B. CDI < 1 = 1 or >1 indicates that the drugs are synergistic, additive or antagonistic, respectively. CDI < 0.7 indicates that the drug is significantly synergistic. p73 activator (RETRA) and NAMPT inhibitor (FK886) act synergistically (CDI = 0.66) to repress cell proliferation in Detroit 562 cells. Another p73 activator (NSC59984) and NAMPT inhibitor (FK886) also have synergic effects (CDI = 0.6) that reduce cell growth.

## Data Availability

Data can be provided from the corresponding author upon reasonable request.

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
