# Peer review of "NAMPT Inhibitor and P73 Activator Represses P53 R175H Mutated HNSCC Cell Proliferation in a Synergistic Manner"

_biomolecules, 2022, doi:10.3390/biom12030438_

Round 1

Reviewer 1 Report

The manuscript ‘NAMPT inhibitor and P73 activator represses p53 R175H mutant HNSCC cell proliferation in a synergistic manner’ by Cai et al examines whether NAMPT inhibitor can act in conjunction with p73 activator to repress growth in Head and Neck Squamous cell cancer (HNSCC) cells.

The p53 family of proteins comprises p53, p63 and p73 which are multiply spliced.  P53 is a tumour suppressor protein that is frequently mutated in HNSCC.  Most p53 mutants are loss of function but some acquire gain of function activities.  It is known that aggregation of mutant p53 can induce GOF.  P73 activators can have an anti-cancer effect in p53 mutant cells but p73 activators are not effective in all HNSCC cell limes.  NAMPT has been shown to be a key regulator of mutant p53 aggregation.  Here they show that a NAMPT inhibitor when used in conjunction with a p73 activator was able to repress growth in HNSCC cells with p53 GoF function mutant, suggesting a potential for combination therapy for HNSCC with p53 GoF mutation. 

The results presented are clearly an interesting observation.  However there are a number of questions:

  1. What is NAMPT? 
  2. The finding presented is for one p53 GoF mutant.  How general is this finding- does it hold up for other GoF mutants?
  3. What defines whether a mutant is GoF?  It’s implied that it is ability to aggregate?  Does this mean to aggregate with p53 itself or the ability of this mutant p53 to heterodimerise with p73 or p63 which I believe is one of the mechanisms how mutant p53 can acquire GoF activities?
  4. Is the reason why some HNSCC cells do not respond to a NAMPT activator is not only because p53 is not a GoF but what if the cells do not express p73 or p63 or another transcription factor or a factor with which mutant p53 interacts to acquire additional oncogenic functions.
  5. Is the ability of mutant p53 to be GoF dependent upon expression of the correct isoform of p63 and/or p63?
  6. I donot fuly understand the results in Fig 3.
  7.  The thioflavin aggregates in cells-would it be correct that these aggregates are not necessarily aggregates of mutant p53 but aggregates of any protein.  

Reviewer 2 Report

The paper describe the response of two head and neck squamous cell carcinomas cell lines, OECM1 and Detroit 562 cells with different inhibitors and activators.

The report is concise and most of the results are clearly presented. However, some of the assumptions and conclusions are not convincing. In line 128 it is stated that "all protein aggregation acquires a new β-chain structure following an aggregation process involving intermolecular interactions", which is completely false. There are all kinds of aggregations. Several false assertions follow in the text from this basic misunderstanding.

ThT is normally used to test for amyloid fibril formation, however in some cases it has been found that THT can bind to molecules that are not forming fibrils.

Furthermore, the paper does not seem to have much to do with the special issue "Unveiling the functions, dynamics and interactions of proteins in health and disease using theoretical and experimental approaches". For instance does not include any of the scopes of the number: molecular biology, structural biology, biophysics, biochemistry, bioinformatics or computational chemistry.

- Line 53: Not clear: “We predicted that the p53 R175H and V173L mutations were prone to aggregation or not, (respectively) so that the idea makes sense”.

- Some names are written differently throughout the text: GOF or GoF, NAMPT inhibitor (FK886), etc

- Line 81: The final concentration of Hoechst 33342 (1 mg/ml) is the same as the initial one?

- Figure 1: Why are Retra and NSC59984 used at different concentrations? One is more cytotoxic than the other? It should be mentioned why such different concentrations.

-Figure 3: Following the same reasoning as the analysis for V173L. Why doesn't R175H have a higher beta sheet CSSP value than WT? Although the "prediction" of secondary structure may provide information, it does not represent a strong evidence for the formation of "aggregates". There are several examples where a very strong beta sheet propensity does not correlate with fiber formation at all. You need a better description to be able to understand the figure: what do each panel describe? colors?.

- Figure 4: ThT staining it seems very similar to that of Hoecht 33342. why? If each one stains different structures, shouldn't the images be more different? There is no prove that THT is binding p53 agregates.

- Figure 6: The y axis need to be renamed. What is really the measure?. Controls of the interaction of ThT with the inhibitors?

Reviewer 3 Report

In this study, Cai et al investigated the growth suppression of HNSCC cell lines with mutant p53 by cotreatment with NAMPT Inhibitor and p73 activator. Their investigation relays on the interaction between mutant p53 and p73. The authors call these interactions aggregates. Once p73 is released from this complex, it has tumor suppressive behavior. However, the treatment with p73 activator is not sufficient to suppress Detroit 562 cell line bearing a p53 R175H mutation. They found the inhibitor of NAMPT to cooperates with p73 activator to suppress  Detroit 562 cell growth.

The research is interesting and well performed. 

However, there are some concerns:

  1. It should be at least mentioned that p73 has several isoforms, some have tumor suppressive behavior, some oncogenic. Which forms complex with mutant p53? The introduction needs some more data.
  2. The Figure 4 has bad quality. It should be improved. Anyway, thioflavin T stains protein aggregates in general, it would be better to confirm that this is complex between mutp53 and p73.
  3. The discussion is poorly written, there is no analysis of the obtained data in the light of existing achievements in the literature. 
  4. The English should be improved.

Round 2

Reviewer 1 Report

The authors have clearly addressed the reviewers comments.  This has led to the paper being much improved.

I have some minor suggestions/edits:

L26/27: NAMPT inhibitor, to reduce abnormal aggregation of mutant p53, used in a co-treatment with a p73 activator was able to effectively repress growth in HNSCC cells with p53 GoF mutant.  Please rephrase

L45: sedimentation- not correct- association or sequesteration

L49: reactive- not correct- reactivate.  Probably worth saying that mutant p53 can associate with p63 and p73 whereas WT p53 does not. 

L56: mutation sites- simper to say mutations

L170-: greem- should be green

L226: sentence does not make sense

L242: organ- should be organs

L269: mutants- should be mutant

Reviewer 2 Report

The authors have addressed my concerns.

Author Response

We have conducted three round of English editing using professional English editor in the life sciences. We thank you for previous comments and suggestions to help us improvement.